# Genome-Wide Identification and Expression Analysis of Heat Shock Transcription Factors in *Camellia sinensis* Under Abiotic Stress

**DOI:** 10.3390/plants14050697

**Published:** 2025-02-24

**Authors:** Guimin Li, Xinying Shi, Qinmin Lin, Mengmeng Lv, Jing Chen, Yingxin Wen, Zhiyi Feng, Syed Muhammad Azam, Yan Cheng, Shucai Wang, Shijiang Cao

**Affiliations:** 1Laboratory of Plant Molecular Genetics & Crop Gene Editing, School of Life Sciences, Linyi University, Linyi 276000, China; liguimin@lyu.edu.cn (G.L.); chenjing2488@outlook.com (J.C.); 2College of Life Sciences, Fujian Agriculture and Forestry University, Fuzhou 350002, China; 13163835793@163.com (X.S.); 18760030926@163.com (Q.L.); 3College of Forestry, Fujian Agriculture and Forestry University, Fuzhou 350002, China; youyosihq@163.com (M.L.); 18050409132@139.com (Y.W.); 4College of Resources and Environment, Fujian Agriculture and Forestry University, Fuzhou 350002, China; savannah11210802@163.com (Z.F.); syedazamfafu@gmail.com (S.M.A.); 5College of Plant Protection, Fujian Agriculture and Forestry University, Fuzhou 350002, China; chengyan1220@hotmail.com

**Keywords:** tea, Hsf, gene family, abiotic stress

## Abstract

The tea plant (*Camellia sinensis*) is an economically important crop that plays an important role not only in the beverage industry but also in the pharmaceutical industry. The environment has a great influence on the quality of the tea plant. Heat shock factors (Hsfs) are transcriptional regulators that control the plant response to adversity. However, only a limited number of studies have reported the *Hsf* gene in *Camellia sinensis*, and most of these reports involve high-temperature, drought, and salt stress. Research on light, dark, and cold stress is limited. In this study, 22 *CsHsf* genes were obtained by whole genome sequencing and found to be located on 11 chromosomes. In addition, the gene structure, protein motif, and phylogeny were studied. We classified the genes into three major subfamilies: CsHsfA, CsHsfB, and CsHsfC. Interestingly, we found that there was more alignment between *CsHsf* and *Hsf* genes in dicotyledons, including *Arabidopsis thaliana* and *Solanum lycopersicum*, than in the monocotyledon *Oryza sativa*. The expression of many *CsHsf* genes was affected by low-temperature, light, and dark abiotic stresses. Notably, *CsHsf15* and *CsHsf16* showed high induction rates under both light and cold stress, and both genes carried *cis*-acting elements associated with light and low-temperature responses. These results lay a solid groundwork for further investigations into the involvement of *CsHsf* genes in the response of *Camellia sinensis* to abiotic stresses.

## 1. Introduction

The tea plant (*Camellia sinensis*) is native to Southeast Asia, East Asia, and the Indian subcontinent, but it is now widely grown in tropical and subtropical regions around the world [1]. As one of the world’s most significant beverage crops, the tea plant (*Camellia sinensis*) has substantial economic and health benefits, making it an essential cash crop globally [2]. However, tea plants are susceptible to a variety of environmental stresses throughout their life cycle. On the one hand, low temperatures under abiotic stress can change the indole formation of tea plants [3], salt stress can cause the accumulation of reactive oxygen species (ROS) [4], and drought stress can significantly affect the biosynthesis of abscisic acid, ethylene, and jasmonic acid and the expression levels of genes related to signal transduction [5]. On the other hand, when herbivores eat tea under biological stress, it affects the yield of tea [6,7].

To cope with these environmental challenges, plants have evolved sophisticated defense mechanisms, with transcription factors playing a critical role in activating or inhibiting the expression of stress-responsive genes. Transcription factors are DNA-binding proteins that can activate or inhibit the transcription of multiple target genes by binding to certain gene promoter regions [4,8,9]. Plants can regulate the expression of related stress-resistance genes at the mRNA level and then cause a series of physiological and biochemical reactions in plants to adapt to the environment [2,10].

Among the various defense mechanisms, heat shock proteins (HSPs) are highly conserved molecular chaperones crucial for the plant defense against heat stress and other abiotic stresses [11,12]. HSPs aid in maintaining cellular structure and function, which is essential for plant survival in stressful environments. At present, the functional mechanisms of HSP90, HSP70, and sHSP in plant stress responses have been widely reported. For instance, the expression of Arabidopsis *HSP90* genes is strongly induced under heat, cold, salinity, and heavy metal stress. The overexpression of alfalfa’s *MsHSP70* and horse gram’s *MuHSP70* can improve the resistance of plants to cold, heat, drought, salinity, and oxidation stress [13] Heat shock factors (*Hsfs*), which are a group of transcription factors, regulate the expression of HSP genes in response to stress. The regulatory capacity of *Hsfs* is largely due to their ability to preserve protein stability under stressful conditions [14,15].

Hsfs have a highly conserved protein structure with five key domains: the nuclear localization signal (NLS), nuclear export signal (NES), DNA-binding domain (DBD), oligomerization domain (OD), and a short C-terminal activator peptide motif (AHA motif) [16,17,18]. The NLS allows Hsf proteins to be transported from the cytoplasm to the nucleus, where they can bind to the heat shock elements in the promoter regions of HSP genes, triggering their transcription [19,20,21]. Hsfs are classified into A, B, and C groups based on the structure of their HR-A/B region, with A-group Hsfs having a more complex structure than the simplified B- and C-group Hsfs [22].

The function of the Hsf gene family has also been reported in previous work. *ZoHSF16* and *ZoHSF25* are believed to play a role in ginger’s response to high-temperature and strong light stress [23]. *AeHSFA2b* significantly enhances the tolerance of *Actinidia eriantha* to salt stress by increasing the expression of *AtRS5*, *AtGolS1***,** and *AtGolS2* [24]. In peanuts (*Arachis hypogaea* L.), the overexpression of *AhHsf20* enhances salt tolerance. In rice (*Oryza sativa* L.) [25], *Os03g53340* is essential for the early activation of heat shock protein genes under heat stress [26]. These results suggest that plants have evolved subclass specificity and multiple functions in some members of the Hsf family. In addition, a large number of studies have shown that the *HSF* family plays an important regulatory role in various physiological aspects of growth and development in other plants. For example, the overexpression of *TaHSFB4*-2B inhibits the seed germination and growth of *Arabidopsis thaliana* under salinity and mannitol treatment [26]. The whole genome sequencing of plants like rye (*Secale cereale* L.), the common bean (*Phaseolus vulgaris*), *Arabidopsis*, rice, and alfalfa (*Medicago sativa*) has further deepened our understanding of these gene families [27,28,29,30,31,32].

Despite extensive research on Hsf genes in various plants, studies on Hsf genes in tea plants remain limited, and most of the existing research has primarily focused on their response to high-temperature, salinity, and drought stress. Previous studies have shown that *CsHsfs* are differentially regulated by drought, salt, and heat stress, and the heterologous expression of *CsHsfA2* in yeast has been found to improve thermotolerance, suggesting its potential role in regulating heat stress responses [33]. Our study, which explores the response of tea trees to light, low-temperature, and dark stress, fills a gap in current knowledge by addressing these less-explored stress factors. Given that abiotic stresses significantly reduce tea plant yield and considering the crucial role of *Hsfs* in helping plants cope with adverse environments, a deeper understanding of *CsHsf* genes in tea trees is essential for advancing future research. In this study, we identified 22 *CsHsf* genes in tea plants and examined their chromosomal locations, *cis*-regulatory elements, phylogenetic relationships, conserved regions, and gene structures. We also explored the intrinsic mechanisms of *CsHsf* genes in the tea plant’s adaptation to stress. The findings of this research provide a valuable theoretical foundation for further investigation into the functions of *Hsf* genes in tea trees and other species as well as for advancing plant genetic improvement.

## 2. Results

### 2.1. Identification of the CsHsf Genes in Camellia sinensis

In this study, we successfully identified 22 *CsHsf* genes from the genome of *Camellia sinensis* ‘Tieguanyin’. These genes were named *CsHsf1* to *CsHsf22* based on phylogenetic relationships (Table 1 and Figure 1). A variety of gene characteristics were identified, including locus name, amino acid sequence length, protein molecular weight, isoelectric points, instability index, aliphatic index, and predicted location (Table 1). In particular, the length of these genomic sequences is variable, with *CsHsf12* containing only 79 amino acids (aa), the shortest of the 22 *Hsf* genes. In contrast, the longest, *CsHsf13*, has 536 amino acids (aa). These 22 genes vary in protein molecular weight from 9.030 kDa (*CsHsf12*) to 60.823 kDa (*CsHsf13*), ranging from 4.75 (*CsHsf15*) to 8.97 (*CsHsf18*) in isoelectric points. The instability index varies between 36.81 (*CsHsf12*) and 80.56 (*CsHsf17*). All 22 members of this family have an instability index higher than 40 and are therefore considered unstable. Of particular note, subcellular localization shows that all members of the *CsHsf* family are localized to the nucleus.

### 2.2. Phylogenetic Analysis and Classification of CsHsfs

We used 22 Hsf proteins to construct a phylogenetic tree with 25 and 26 Hsf proteins from Arabidopsis thaliana and Solanum lycopersicum. The results show that members of the CsHsf family are found in all three groups. According to the results, we divided Hsf into three groups: HsfA, HsfB, and HsfC. The largest of these is CsHsf A, consisting of 10 proteins (CsHsf11 CsHsf12, CsHsf1, CsHsf7, CsHsf21, CsHsf4, CsHsf16, CsHsf18, CsHsf13, CsHsf9), accounting for 45.5% of the total CsHsfs. It was followed by CsHsf B, accounting for 40.1%. CsHsf C had the least protein members, with only three proteins, namely CsHsf15, CsHsf10, and CsHsf20. Phylogenetic tree studies show that the tea plant is closely related to Arabidopsis thaliana and Solanum lycopersicum.

### 2.3. Distribution of CsHsf Genes Across Chromosomes

We identified 22 *CsHsf* genes in the tea plant genome, and the 22 identified sequences were more uniformly localized on 11 chromosomes of the tea plant (Figure 2). To avoid confusion, we named these sequences *CsHsf01* to *CsHsf22*. The *CsHsf* genes were distributed across all the chromosomes, with the highest number of *CsHsf* genes distributed on chromosomes 1 and 14. The lowest number of *CsHsf* genes were distributed on chromosomes 4 and 13, all of which had one *CsHsf* gene.

### 2.4. Gene Structure and Conserved Motif Composition

To further elucidate the structural features of the 22 Hsf transcription factor families, we thoroughly analyzed its exons and introns, focusing on their quantity and distribution (Figure 3D). The gene structure analysis revealed that the number of exons in different CsHsf genes varied from two to five. Most of the CsHsf genes had two exons. There were five CsHsf genes (CsHsf12, CsHsf21, CsHsf10, CsHsf20, and CsHsf22), and only CsHsf2 contained one exon. Moreover, members of genes from the same family show similar exon and intron structures.

Using the online MEME tool, we conducted a further analysis of conserved motifs within the Hsf protein family members. The results showed that motifs 1, 2, and 3 were present in all the members except CsHsf12 and CsHsf21, although CsHsf12 contained only motif 2 and CsHsf21 only motifs 1 and 2 (Figure 3B). Notably, motif 8 and motif 6 were found only in group A Hsfs, and motif 9 was found only in group B Hsfs. Overall, each group of CsHsf members shared similar motifs, yet notable differences were observed between the various groups (Figure 3B).

### 2.5. Analysis of cis-Acting Elements

In order to deeply investigate the response of *CsHsf* genes to abiotic stresses, we used preliminary experimental validation and data analysis (Figure 4). We found that *CsHsfs* contain a series of cis-regulatory elements, stress-related elements (anaerobic induction, low-temperature responsiveness, anoxic specific inducibility, and defense and street responsiveness), hormone-related elements (salicylic acid response, abscisic acid response, MeJA response), and development-related components (light responsiveness, meristem expression, control of circadian rhythms). The results showed that there were 18 genes with *cis*-acting elements related to light response, 8 genes with *cis*-acting elements related to low-temperature response, and 12 genes with *cis*-acting elements related to the abscisic acid reaction.

### 2.6. Intraspecific Collinearity Analysis of CsHsf Genes

The main causes of gene function changes are the loss and gain of genes [8]. To investigate the replication events of *Hsfs*, we conducted a collinear analysis within the species (Figure 5). The analysis results indicate that 10 genes out of the 22 *CsHsfs* have collinear relationships, with a total of 7 gene pairs. We also found that some *CsHsf* genes were linearly related to multiple genes, such as *CsHsfs16* being collinear with both *CsHsfs18* and *CsHsfs4*. Combined with the evolutionary analysis, all three genes belonging to taxon A. *CsHsfs19* were collinear with *CsHsfs2* and *CsHsfs3*, all of which exist in the B taxon. This indicates the tendency of genes with collinearity to be clustered in the same taxon. These genes may arise through genome replication events and are structurally and functionally similar. In addition, 10 *CsHsfs* were involved in segmental duplication, and the gene duplication events all took place between different chromosomes, which suggests that segmental duplication is the main driving force in the evolution of *CsHsfs*.

### 2.7. Synteny Analysis Among CsHsf Genes

In order to explore the underlying evolutionary mechanisms driving the development of tea plants, we conducted a comparative analysis of collinearity among *CsHsf* gene pairs in the genomes of *Arabidopsis thaliana*, *Solanum lycopersicum*, and *Oryza sativa* (Figure 6). We found that 16 homologous gene pairs were identified between *CsHsf* and *AtHsf* genes, 10 homologous gene pairs were found between *CsHsf* and *OsHsf* genes, and 29 homologous gene pairs were found between *CsHsf* and *SiHsf* genes. We found more collocated genes between *CsHsf* and *Hsf* genes in the dicotyledonous plants *Arabidopsis thaliana* and *Solanum lycopersicum* than in the monocotyledonous plant *Oryza sativa*. In addition, three *CsHsf* genes, *CsHsf5*, *CsHsf2*, and *CsHsf19*, showed collinearity with the *Arabidopsis thaliana*, *Oryza sativa*, and *Solanum lycopersicum* genes. Based on the collinearity analysis, it can be inferred that these three *CsHsf* genes likely existed prior to the divergence of these species.

### 2.8. Expression Patterns of CsHsfs Under Light, Shade, and Cold Stress

We used quantitative real-time PCR (qRT-PCR) to confirm how the *CsHsf* gene responds to different abiotic stresses, including light, shade, and cold. The results showed that the expression levels of all *CsHsf* genes increased and decreased significantly under different stress treatments (Figure 7). Under light stress, *CsHsf* gene expression showed a trend of first increasing, then decreasing, and then increasing, increasing at 4 h, downregulating at 8 h, and upregulating again at 12 h and 24 h. *CsHsf16* had the most significant upregulation, followed by *CsHsf22*, *CsHsf8*, and *CsHsf15*. Most *CsHsfs* peaked at 4 h and decreased significantly at 8 h, indicating that the hsf family has a certain response to light stress. Under shade stress, the expression of *CsHsf* genes was suppressed to varying degrees. All *CsHsfs* showed a downward trend at 4 h and then upregulated under shade stress, reaching a peak at 24 h. This trend may reflect the potential role of these genes in plant adaptation to low-light conditions. Under cold stress, *CsHsf* gene expression decreased at 4 h and 12 h and increased at 8 h and 24 h. It is worth noting that the expression level of *CsHsf15* increased the most significantly, and the expression level increased by about six times compared with the initial stage of treatment. This was followed by *CsHsf8*, *CsHsf16*, and *CsHsf17*. This may indicate that *CsHsf* family members have a potential ability to withstand cold stress.

## 3. Discussion

Changes in global temperatures have caused severe abiotic stress, and a growing body of research supports the critical role of the Hsf gene in plants coping with harsh conditions [25]. Verifying the function of plant genes can be accomplished effectively by plant genome analysis [34]. The *Hsf* gene has been studied in a variety of plants in *Hsf* gene family members. These include garlic [35], alfalfa [30], soybean [36], wheat [37], *A. thaliana* [38], *Vitis vinifera* (grapes), and *Beta vulgaris* (sugar beet). However, until now, only a limited number of studies have been conducted on the *Hsf* gene in tea plants. In the limited number of studies, most have involved the response of tea *Hsfs* to high-temperature, drought, and salt stress. In order to improve the comprehensive understanding of the tea Hsf gene family, Liu et al. [39] first identified the *Hsf* gene family in *Camellia sinensis* using RNA-seq data. However, limited by the lack of reference genome sequencing, only 16 *CsHsf* genes were identified. Here, we used *Camellia sinensis* as the research object to identify the *CsHsf* family using high-quality tea tree reference genomes. In our study, the *Hsf* gene family of tea trees was studied in depth, and 22 *CsHsf* gene family member encoding transcription factors were identified. There were less *Hsf* genes in tea plants than in rice (25) [40], populus (28), and soybeans (26) but more than in algae. Previous studies have shown that there are certain differences in the content of Hsf transcription factors in different species, and the content of Hsf in land plants is significantly higher than that in algae, which is consistent with our research results.

Subsequently, the basic physicochemical properties of CsHsfs were analyzed, as shown in Table 1. Nineteen of the 22 members of the CsHsf protein are unstable in vitro, which is consistent with most studies showing that most members of the Hsf gene family are behaviorally unstable proteins [30]. From a biological significance perspective, unstable proteins may be more easily recognized and regulated by intracellular regulatory mechanisms. When plants respond to environmental stresses, this instability may help cells rapidly adjust the content of Hsf proteins. The isoelectric points of the Hsf gene family exhibited similarities. The theoretical isoelectric points of CsHsfs range from 4.75 to 8.97, which is very similar to those of *Pisum sativum* [41]. All CsHsfs are distributed in the nucleus, which may be an activator that induces the expression of other genes.

To elucidate the evolutionary links between *Arabidopsis thaliana*, *Solanum lycopersicum* L., and *Camellia sinensis*, we constructed a phylogenetic tree of Hsfs. Phylogenetic analyses showed that CsHsfs could be classified into three major groups, corresponding to the Hsfs of *Arabidopsis thaliana* and *Solanum lycopersicum* L. This result is in agreement with previously reported results [33]. Subfamily Hsf A had the most members, and subfamily Hsf C had the least members. The results showed that *Camellia sinensis* had three fewer Hsf members than *Arabidopsis thaliana* and four fewer than *Solanum lycopersicum* L. *Camellia sinensis* has fewer Hsf members than other plant species, which may reflect the insufficient expansion of the CsHsf family. Interestingly, the Hsf family members of these three species were not isolated but showed a cross-distribution pattern (Figure 1). This suggests that the evolution of the Hsf family is more conserved in dicot plants. Since genes clustered in the same subpopulation may handle similar functions, we can infer the function of our gene family by clustering nearby *Hsfs*. Notably, *CsHsfs* frequently congregate with *Arabidopsis thaliana*. The *Hsf* gene of *Arabidopsis thaliana* can increase cold resistance by activating the expression of the Hsp20 family of proteins when induced by cold stress. The aggregation of *CsHsfs* with *AtHsfs* may indicate a similar function for *CsHsfs* [42].

In order to fully understand the *Hsf* gene family, we further studied the motif, domain structure, and gene structure of the CsHsf protein. Short sequences that are important for biological processes are made into motifs. During evolution, members within the same genus and species have similar genetic structures and closely related motifs [43]. In this experiment, in terms of protein motifs, most CsHsfs have one to four motifs, which may indicate that they have a general role in biology. Motif 6 was identified only in group CsHsf A, and motif 9 was found only in group CsHsf B, which may indicate that these two sets of Hsf genes have unique functions. In addition, in terms of gene structure, we found that the same group of *CsHsf* genes had the same intron–exon structure (Figure 3). In the present study, there were two to five exons in the *CsHsf* genes. Previous research has revealed that *Medicago sativa* L. *Hsf* genes contain two to three exons [44]. In the maize Hsf gene family, the number of exons ranges from one to three [45], and this, to a certain extent, reflects the evolutionary differences between dicotyledonous plants and monocotyledonous plants from the perspective of gene structure.

Unlike animals, plants are constantly exposed to environmental stress, which leads them to evolve a range of molecular mechanisms to adapt to environmental changes. The *cis*-element of the promoter plays a key role in initiating gene expression [46]. Through the analysis of promoters, a large number of core promoter elements and binding sites related to hormone response and stress are distributed in the 22 Hsf promoter regions of *Camellia sinensis*. Notably, most of the *CsHsf* members have abscisic acid reaction-associated elements, suggesting that *CsHsfs* may be involved in communication between signaling pathways in abscisic acid reactions. These signaling pathways also include a wide range of calcium ion-related signaling pathways.

Based on our results, we selected nine *CsHsf* genes from different subfamilies and analyzed their expression patterns. The expression of *CsHsf* genes under light stress, shade stress, and cold stress was analyzed by qRT-PCR. The results showed that the expression of most *CsHsf* genes was downregulated, and some genes were upregulated under cold stress. This expression pattern was consistent with those of *Dianthus caryophyllu* and *Verbena bonariensis* [46,47]. Notably, *CsHsf15* and *CsHsf16* showed high induction rates under light and cold stress, and both genes carried *cis*-acting elements associated with light and low-temperature responses. This may indicate that these genes play a role in stress resistance during the development of *Camellia sinensis*. However, due to the lack of functional verification experiments, the specific regulatory mechanisms cannot be determined at present. Interestingly, in a previous study, the *CsHsfB1* gene was highly upregulated under cold treatment, and its expression was increased 3.3- and 4.3-fold in ‘Yingshuang’ and ‘Anjibaicha’ at 24 h and 4.7-fold in ‘Yunnanshilixiang’ [39]. In contrast, in our study, the expression of *CsHsf15* and *CsHsf16* increased threefold and onefold at 8 h, respectively. The time-specificity of the increase in expression under cold treatment in different varieties of *Camellia sinensis* may reflect the variability in the response of different genes. However, *CsHsf15* and *CsHsf16* can be induced by light, showing a trend of first increasing and then decreasing, which is consistent with the expression patterns of *ZoHSF05*, *ZoHSF12*, *ZoHSF16*, and *ZoHSF25* in ginger [23]. Our study has played a role in mutual validation with the *ZoHSF* gene family in ginger.

Hsf plays an important role in the plant response to abiotic stress [48]. Plants can mitigate the effects of abiotic stress by modulating the expression of various *Hsf* genes. Upon sensing environmental stress signals, such as low temperature, light changes, or darkness, plants activate downstream signaling pathways via Ca^2^⁺ receptors to regulate their adaptation to adversity. When Ca^2+^ binds to calmodulin (CaM) [49], it induces the expression of key genes like CDPK, initiating multiple stress tolerance pathways [50]. Previous studies have shown that specific types of transcription factors of HSF not only regulate the expression of HSP genes but are also involved in interactions between HSP proteins [51,52]. All of us hypothesized that *HSF* first expresses HSP proteins before initiating the regulation of the expression of core enzymes in the lipid metabolic pathway (e.g., *GPAT*, *SAD*, *KCS*), followed by the promotion of the synthesis of key lipids, such as phosphatidic acid (PA), phosphatidylethanolamine (PE), and digalactosyldiacylglycerol (DGDG), thereby enhancing cell membrane stability and increasing cold resistance [53]. Under stress, the expression of genes related to jasmonate (MeJA) synthesis is suppressed, leading to reduced MeJA pathway activity, accompanied by malondialdehyde (MDA) accumulation and electrolyte leakage, indicating cellular damage [54]. Our study found that some genes involved in the stress response contain MeJA response elements, and the presence of these elements is closely related to the sensitivity of genes to stress signals. The expression of the *CsHsf* gene promotes the accumulation of ABA, which in turn promotes the synthesis of MeJA, thereby increasing the activity of antioxidant enzymes. The accumulation of MeJA also increases the production of jasmonic acid (JA), and all these series of changes significantly improve the plant’s stress tolerance [55]. In our study of cis-elements (Figure 4), we found a potential relationship between *CsHsfs* and these key response proteins, suggesting that *CsHsfs* may be resistant to environmental stress through the interaction of these proteins. Plants produce toxic reactive oxygen species (ROS) in response to different types of stresses, and the excessive accumulation of ROS is harmful to cells. Therefore, the timely removal of ROS becomes crucial for plants to enhance their stress tolerance. With the rising MeJA content, the antioxidant system of plants is activated and the ROS scavenging rate improves [56]. Plants demonstrate their multi-level adaptive capacity and physiological regulation potential by maintaining stable growth under complex environmental stresses through multiple defense mechanisms in vivo, such as signal transduction, metabolic regulation, and gene expression regulation (Figure 8). This mechanism is derived from our research on *CsHsfs* and other hypotheses about plant responses to stress. More experiments are needed to confirm whether *CsHsfs* help plants cope with stress challenges by regulating the expression of related proteins.

## 4. Materials and Methods

### 4.1. Identification and Characterization of the Hsf Gene Family in Camellia sinensis

The genome sequence and corresponding annotation data for *Camellia sinensis*, sourced from the Tea Research Institute of the Fujian Academy of Agricultural Sciences, were obtained through PacBio sequencing technology, followed by a de novo assembly process. Leaf samples were collected from a solitary tea plant (*Camellia sinensis*), cultivated in Anxi County, Fujian Province, China, located at coordinates 119.576708° E, 27.215297° N. To identify *Hsf* genes, we conducted a hidden Markov model (HMM) analysis against the conserved domains (PF00447) using the Pfam protein family database (http://pfam.xfam.org/, accessed on 2 June 2024). The Hummer search tool (version 3.0) was utilized to pinpoint genes containing these domains. Further validation of the CsHsf gene family was achieved by aligning the amino acid sequences of Hsfs from Arabidopsis thaliana, which were retrieved from the PlantTFDB database (https://planttfdb.gao-lab.org/, accessed on 2 July 2024) by utilizing the BLASTp tool available at NCBI. The structural domains were analyzed utilizing the NCBI-CDD search tool (found at https://www.ncbi.nlm.nih.gov/Structure/bwrpsb/bwrpsb.cgi, accessed on 2 July 2024) and the Pfam network database (available at https://pfam-legacy.xfam.org/, accessed on 2 July 2024). A total of 22 genes were ultimately identified and designated as CsHsf1-22.

### 4.2. Evolutionary Analysis and Gene Structure of the CsHsfs

To identify conserved motifs within the CsHsfs, the MEME website (https://meme-suite.org/meme/tools/meme, accessed on 3 July 2024) was employed, configured to search for 10 patterns while maintaining the default settings. The protein sequences were analyzed using the NCBI Batch Web CD-Search tool (accessible at https://www.ncbi.nlm.nih.gov/, accessed on 3 July 2024) to predict their conserved domains. The gene structure, including its introns and exons, was depicted through the Gene Structure Display Server (GSDS2.0; https://gsds.gao-lab.org/, accessed on 6 January 2025).

### 4.3. Physicochemical Characteristics and Subcellular Localization

The physicochemical characteristics of members of the CsHsf family were assessed utilizing the ProParam tool available on the Expasy online software platform (https://www.expasy.org/; accessed on 2 July 2024). The subcellular localization of these proteins was forecasted using Plant-mPLoc (https://www.csbio.sjtu.edu.cn/bioinf/plant-multi/; accessed on 2 July 2024).

### 4.4. Phylogenetic Analysis

The protein sequences for Hsfs derived from Arabidopsis thaliana and tomato were procured from PlantTFDB (https://planttfdb.gao-lab.org/, accessed on 2 July 2024). The phylogenetic tree was built utilizing MEGA software (version 11) by aligning sequences via ClustalW and applying the neighbor-joining method with the JTT + G model, including 1000 bootstrap repetitions. The visual depiction of the tree was enhanced utilizing the iTOL web-based tool (https://itol.embl.de/, accessed on 3 July 2024).

### 4.5. Collinearity and Repetition Analysis

The homologous relationships between CsHsf genes and Hsf genes in Arabidopsis, tomato, and rice were analyzed using MCScanX. The study identified segmental and tandem duplication events in the CsHsf genes. These results were subsequently visualized with the aid of TBtools-II v2.136.

### 4.6. Abiotic Stress Treatments

Tea plant (*Camellia sinensis*) seedlings of uniform size were segregated into two groups: a control group and a stress treatment group. The control group consisted of 30 seedlings, while the stress group contained 3. The stressor conditions included exposure to simulated high-intensity light, total darkness, and a low-temperature environment of 4 °C, all maintained at a relative humidity of 75%. Sample collections occurred at 0, 4, 8, 12, and 24 h post-treatment. Each sample was immediately submerged in liquid nitrogen and stored at −80 °C prior to RNA extraction.

### 4.7. RNA Extraction and Quantitative Analysis

RNA was isolated from both the control and stressed samples using an RNA extraction kit manufactured by Omega Bio-Tek, located in Shanghai, China. Following this, cDNA was produced through the EasyScript one-step process for gDNA removal and cDNA synthesis, utilizing SuperMix provided by Transgen, Beijing, China. A quantitative real-time polymerase chain reaction (qRT-PCR) was carried out with TransStart Top Green qPCR SuperMix from Transgen, Beijing, China, strictly adhering to the manufacturer’s instructions. The reaction mixture was composed of 1 μL of cDNA, 2 μL of primers designed for the specific target, 10 μL of SYBR Premix Ex TaqTM II, and 7 μL of ddH_2_O, which is nuclease-free. The internal reference gene was selected as CsGAPDH (accession no. GE651107). The qRT-PCR thermal protocol involved an initial denaturation step at 95 °C for 30 s, followed by 40 cycles of denaturation at 95 °C for 5 s, annealing at 60 °C for 30 s, and a melt curve analysis comprising 95 °C for 5 s, 60 °C for 60 s, and 50 °C for 30 s. The 2^−ΔΔCT^ method was applied to calculate relative gene expression levels, and statistical analysis was performed utilizing GraphPad Prism version 9.0, accessible at https://www.graphpad.com/, accessed on 20 August 2024. To ensure the reproducibility of the qRT-PCR results, each experiment was conducted using three biological replicates and three technical replicates.

## 5. Conclusions

In this study, the *Hsf* gene family of *Camellia sinensis* was comprehensively and systematically analyzed. A total of 22 members of the *Hsf* gene family were identified, and the analysis covered the physical and chemical characteristics of the proteins, conserved structural motifs, evolutionary relationships, chromosomal distribution, cis-acting element distribution, and synteny. The response of *CsHsf* genes in *Camellia sinensis* to abiotic stresses was investigated. Hsf transcription factors may interact with other signal transduction molecules, such as the HSP protein family, to participate in the resistance response. Further studies showed that *CsHsf15* and *CsHsf16* had high induction rates under both light and cold stress, and both genes carried *cis*-acting elements related to light and low-temperature responses, suggesting that they may play a potential role in light and low-temperature stress resistance. Our results lay the foundation for exploring the function of the *CsHsf* gene family in tea plant growth and development and in stress response. Additionally, they offer valuable insights for enhancing the adaptability of tea plants to extreme climatic conditions.

## Figures and Tables

**Figure 1 plants-14-00697-f001:**
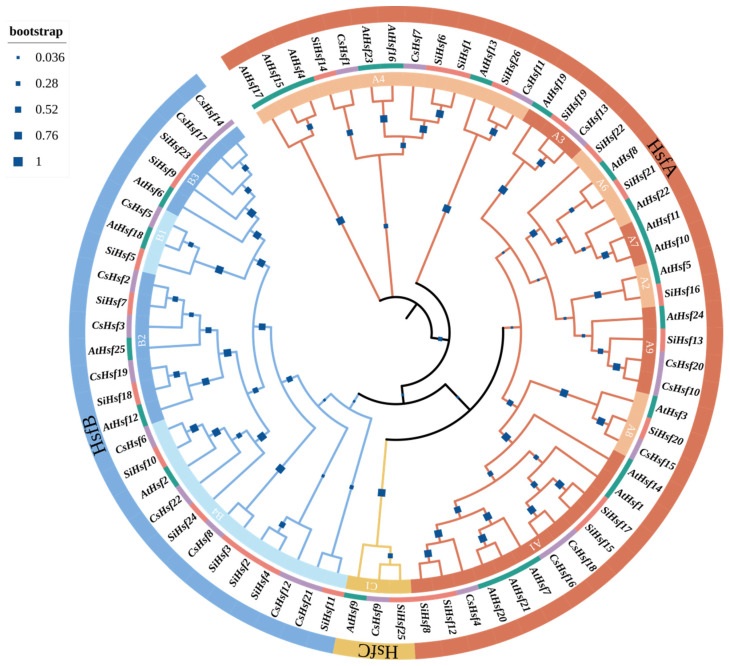
Phylogenetic tree of the *Camellia sinensis*, *Solanum lycopersicum*, and *Arabidopsis thaliana* Hsf family. A phylogenetic tree was generated based on maximum likelihood (ML) analysis, utilizing 1000 bootstrap replicates for accuracy.

**Figure 2 plants-14-00697-f002:**
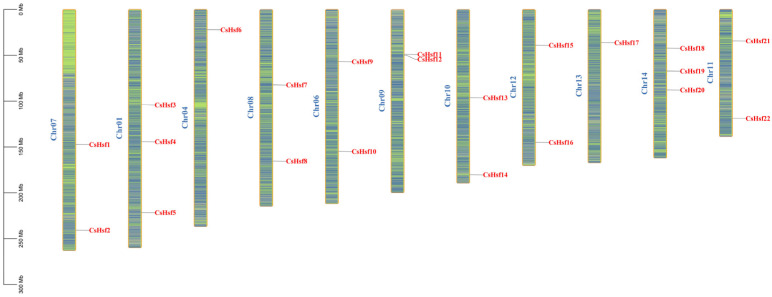
The distribution of *Hsf* genes across the chromosomes in *Camellia sinensis* shows areas of higher gene density marked in blue, while regions with lower gene density are represented in green.The blue letters identify the different chromosomes.

**Figure 3 plants-14-00697-f003:**
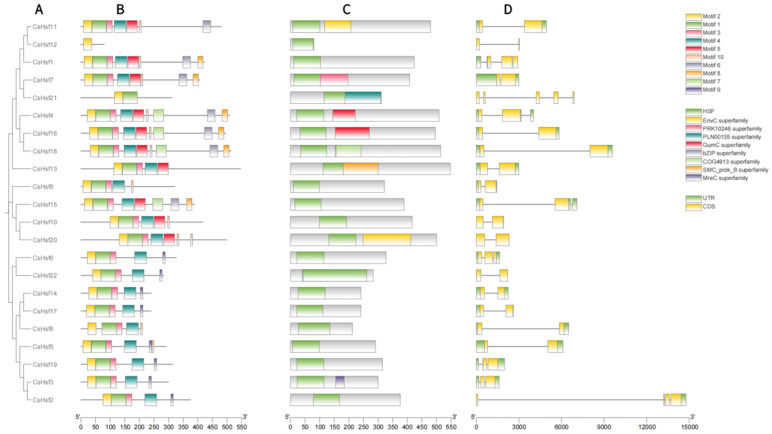
The phylogenetic relationships, conserved protein motif structures, *cis*-acting elements, and gene structure analysis of the CsHsfs are shown. (**A**) A phylogenetic tree was generated in MEGA with the maximum likelihood method, using 1000 bootstrap replicates. (**B**) The distribution of conserved motifs in CsHsf proteins is illustrated, with each of the 10 motifs represented by distinct colored boxes. (**C**) Different color blocks represent different domains. (**D**) The exon–intron structure of the *Camellia sinensis Hsf* gene; exons are indicated by yellow boxes, untranslated regions are indicated by green boxes, and black lines indicate introns.

**Figure 4 plants-14-00697-f004:**
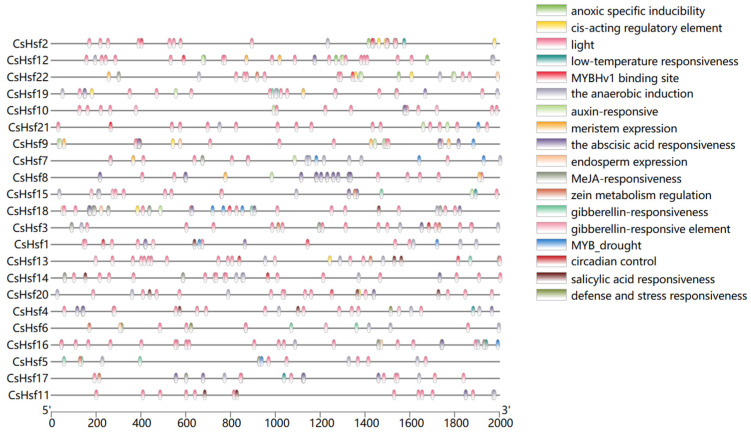
Different colored squares indicate different *cis*-acting elements of the CsHsf promoter region.

**Figure 5 plants-14-00697-f005:**
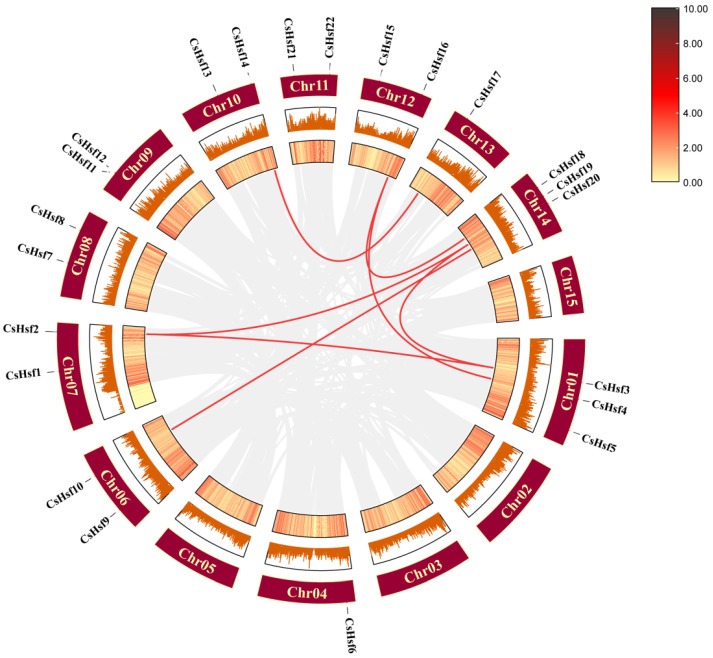
Collinearity among *CsHsf* members in *Camellia sinensis*. Red lines represent collinear relationships among *CsHsf* members, and the gray line indicates all duplicate gene pairs.

**Figure 6 plants-14-00697-f006:**
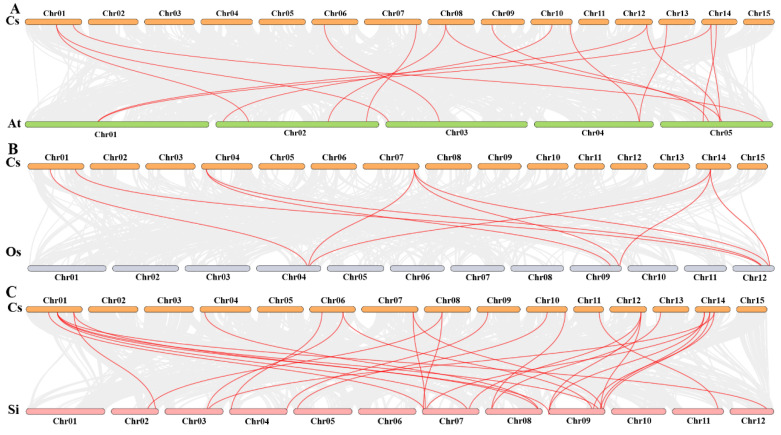
Collinearity analysis plot of *Camellia sinensis* with *Arabidopsis thaliana*, *Oryza sativa*, and *Solanum lycopersicum*. Subfigure (**A**) shows *Camellia sinensis* and *Arabidopsis thaliana*. Subfigure (**B**) is *Camellia sinensis* and *Oryza sativa*, and Subfigure (**C**) is *Camellia sinensis* and *Solanum lycopersicum*. Collinear blocks are marked with gray lines, while collinear gene pairs with *Hsf* genes are highlighted with red lines.

**Figure 7 plants-14-00697-f007:**
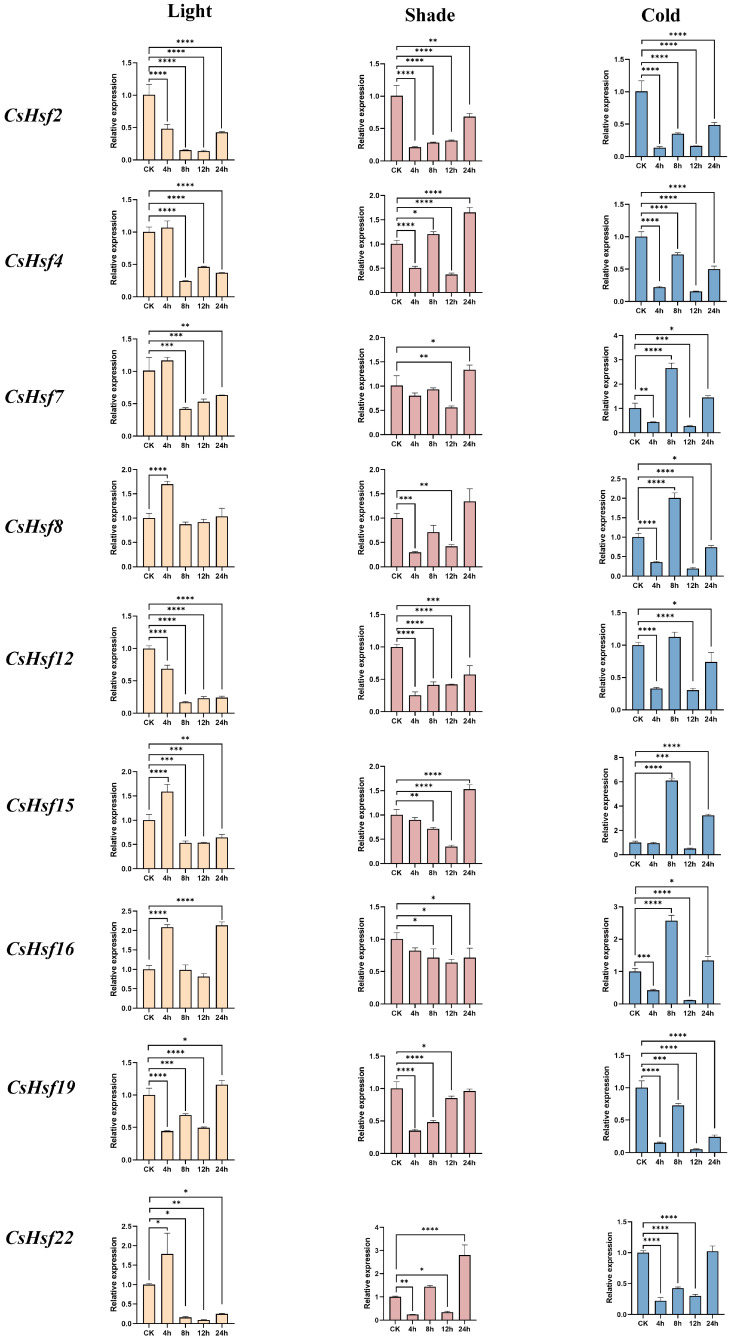
qRT-PCR analysis of *CsHsf* genes under abiotic stress. Light, shade, and cold stress conditions were used for 4 h, 8 h, 12 h, and 24 h, respectively. The data were means ± SD of three biological repeats.* *p* ≤ 0.05; ** *p* ≤ 0.005; *** *p* ≤ 0.0005; **** *p* ≤ 0.0001.

**Figure 8 plants-14-00697-f008:**
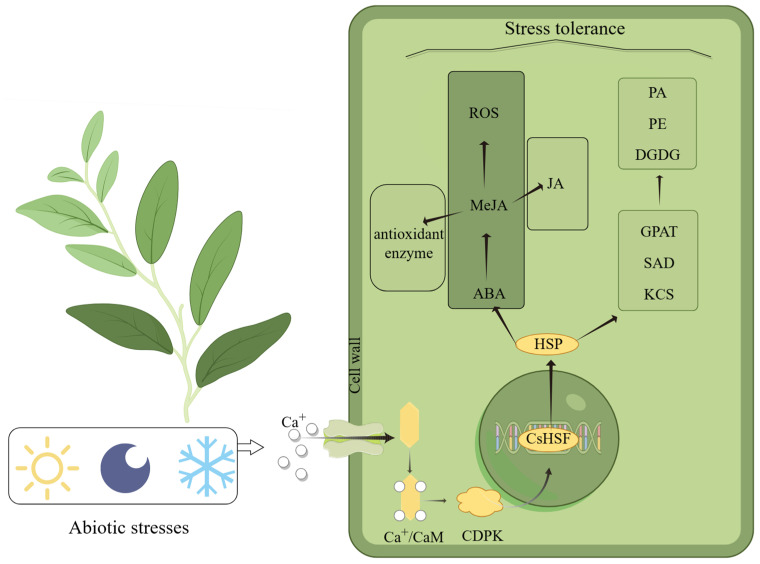
Diagram of the predicted mechanism by which the Hsf gene enables plants to cope with severe environmental stress, produced by Figdraw.

**Table 1 plants-14-00697-t001:** List of the 22 CsHsf genes identified in this study.

Gene Name	Number of Amino Acids/aa	Molecular Weight/Da	Theoretical PI	Grand Average of Hydropathicity	Instability Index	Aliphatic Index	Subcellular Localization
*CsHsf1*	423	48.81	5.28	−0.852	56.04	64.23	Nucleus.
*CsHsf2*	375	41.89	5.88	−0.562	62.11	74.16	Nucleus.
*CsHsf3*	299	34.05	6.47	−0.873	60.21	64.25	Nucleus.
*CsHsf4*	508	56.45	5.13	−0.553	57.85	71.14	Nucleus.
*CsHsf5*	290	32.14	5.38	−0.821	38.90	59.21	Nucleus.
*CsHsf6*	326	36.48	7.71	−0.500	54.80	70.58	Nucleus.
*CsHsf7*	407	46.42	4.98	−0.774	55.45	71.35	Nucleus.
*CsHsf8*	211	24.52	8.97	−0.877	68.00	63.36	Nucleus.
*CsHsf9*	321	36.304	5.75	−0.533	66.85	68.35	Nucleus.
*CsHsf10*	416	46.937	4.95	−0.647	54.37	75.22	Nucleus.
*CsHsf11*	479	53.663	5.76	−0.772	56.17	64.36	Nucleus.
*CsHsf12*	79	9.030	5.07	−0.397	36.81	71.52	Nucleus.
*CsHsf13*	546	60.823	4.95	−0.563	58.24	75.31	Nucleus.
*CsHsf14*	240	27.854	8.20	−0.715	53.05	64.17	Nucleus.
*CsHsf15*	388	44.338	4.75	−0.633	54.97	79.33	Nucleus.
*CsHsf16*	495	54.879	4.89	−0.600	62.72	70.69	Nucleus.
*CsHsf17*	240	27.825	5.60	−0.851	80.56	62.12	Nucleus.
*CsHsf18*	513	56.783	4.78	−0.620	55.54	68.07	Nucleus.
*CsHsf19*	314	35.057	5.37	−0.680	55.13	71.46	Nucleus.
*CsHsf20*	499	55.592	5.15	−0.567	54.76	81.66	Nucleus.
*CsHsf21*	310	35.134	6.46	−0.332	37.59	88.03	Nucleus.
*CsHsf22*	282	32.796	6.39	−0.720	62.81	65.32	Nucleus.

## Data Availability

The data utilized in this study are readily accessible, housed in a publicly available repository. Specifically, the original datasets can be located within the National Center for Biotechnology Information, under accession number JAFLEL000000000, and within the GWH (https://bigd.big.ac.cn/gwh/, accessed on 15 February 2025), where they are assigned the accession numbers GWHASIV00000000.

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
