# Peer review of "Genome-Wide Identification and Expression Analysis of Heat Shock Transcription Factors in Camellia sinensis Under Abiotic Stress"

_plants, 2025, doi:10.3390/plants14050697_

Round 1
Reviewer 1 Report
Comments and Suggestions for Authors
It is a pretty standard review of the expression and organization of HSPs in tea plant. However, there is little new data that is not already covered in multiple other very similar articles. Otherwise, it is well written and nicely illustrated, just redundant with other similar studies. If the authors could highlight how their findings differ from the multitude of previous such articles and show something novel, then it would be acceptable in my opinion. However, as is, it is too redundant with previous such studies to be publishable. I know China limits google, but a better literature review should be done.

Reviewer 2 Report
Comments and Suggestions for Authors
Dear Editor,
I am pleased to have reviewed the manuscript titled "Genome-Wide Identification and Expression Analysis of Heat Shock Transcription Factors in Camellia sinensis under Abiotic Stress." The manuscript is interesting and relevant to a broad audience in the field of stress-related research. These investigations are significant for plant genetic improvement, as they lay the groundwork for future studies on gene insertion at both the nuclear and chloroplast levels.
Abstract
The abstract is engaging and coherently structured. However, it contains typographical errors that should be corrected, such as: "(Camellia sinensis )," where the parentheses are italicized and include extra space. These types of errors should be addressed throughout the document.
Introduction
The introduction is interesting, and the authors attempt to establish a research context. However, this section still requires additional information. Although it includes relevant content, it remains too superficial, making it difficult to establish a strong foundation for the study.
- In scientific nomenclature, the descriptor (e.g., "L.") following a species name should not be italicized. Additionally, in line 44, the scientific name is presented without the descriptor, while in line 46, it is included. This inconsistency should be corrected.
- Lines 47 to 54 contain redundant information, repeating the same ideas. It would be advisable to merge them into a single, more cohesive paragraph explaining the specific effects of different stressors on the plant. This will help the reader better understand their impact.
- Lines 55 to 58 lack scientific depth; restructuring this section is recommended.
- The introduction contains interesting ideas, but there is no logical sequence connecting paragraphs, making it difficult to follow the argument.
I recommend incorporating additional information to strengthen this section by providing examples of relevant genes, detailing the effects of stress, paraphrasing unclear sections, and correcting typographical errors.
Materials and Methods
The methodology is detailed but contains omissions, inconsistencies, and areas that require further clarification or justification:
- The manuscript does not specify how gene expression data were normalized. Which reference gene was used?
- The authors do not explain why the specific time points were chosen for stress treatments.
- Was gene identification validated using another tool or database?
Results
The results are interesting, but as in other sections, it is essential to ensure consistency in formatting and writing style.
- In line 101, some examples are mentioned, but not all start with uppercase letters. This inconsistency is also present in the Discussion section.
- Some conclusions lack strong supporting evidence. For instance:
- "The expression of CsHsf15 increased about 6-fold compared with the expression level at the beginning of the treatment, and it can be hypothesized that CsHsf15 is crucial for the response of Camellia sinensis to low-temperature stress." However, increased gene expression alone is insufficient evidence of function. To confirm this, results should be correlated with previous findings in other species where function has been established through mutant analysis or overexpression/silencing experiments.
Discussion
- The discussion is a critical part of the research as it compares the findings with previous studies. However, I find weak connections between the results and the existing literature. The authors do not clarify whether their results align with or contradict previous findings on Hsfs in plants.
- The conclusions are not sufficiently justified. For example: "Our results will lay the foundation for exploring the function of the CsHsf gene family in tea plant growth and development and stress response." While this is an interesting claim, the study did not experimentally verify gene function; it only reported expression levels.
- Although every study has limitations, this manuscript does not acknowledge any. Identifying these limitations would be beneficial for researchers who may wish to replicate the experiment.
- The manuscript discusses protein instability, but like other aspects, it does not explore the biological significance of these findings. Expanding the interpretation of results would improve the manuscript’s depth.
Conclusions
No comments.
References
The references need to be standardized.
The research findings presented in this manuscript are interesting; however, the document still lacks depth in the interpretation of results. Although comparisons with previous studies are made, the biological significance of the findings is not clearly articulated. Additionally, some hypotheses are not strongly supported by the obtained results. While the manuscript requires substantial improvements, these issues are primarily methodological and argumentative rather than experimental. A thorough revision addressing these deficiencies could make the manuscript suitable for publication.

The document could benefit from an English review to improve the clarity and expression of ideas.
Round 2
Reviewer 1 Report
Comments and Suggestions for Authors
Despite my primary criticism being that this work is redundant as several other very similar studies have been done before on the gene family and in the same species, and even though I provided the citations for those publications, the authors still say there are no other studies. All I was asking for the authors to do was cite the available literature, and describe how their study added to, and compared with, the previous studies. This is the only reason I suggested reject. They have not done so and indeed reiterated the false claim in the introduction and the discussion. I do not work in this field, and only tangentially on tea plant at all, but it was easy to find the other similar paper with a simple google scholar search.
Reviewer 2 Report
Comments and Suggestions for Authors
Dear Editor,
I am pleased with the revisions the authors have made to the manuscript, addressing all the suggestions from my previous review. The manuscript has significantly improved in clarity and presentation, and I consider it suitable for publication in its current form.
Author Response
我衷心感谢您花时间审阅我们的手稿并提供宝贵的反馈。我们非常高兴地听到您对稿件的修改和改进感到满意。您的意见和建议对提高论文的清晰度、结构和整体质量起到了至关重要的作用。
我们非常感谢您为审查我们的工作所投入的时间和精力。您周到的指导帮助我们解决了关键领域,提高了我们研究的准确性和严谨性。您的每一个建议都经过仔细考虑和实施,您的审阅无疑提升了这份手稿的学术水平。
再次感谢您一直以来的支持以及您的反馈对这项工作做出的重大贡献。我们期待您的最终批准,并希望手稿能达到您对出版的期望。
Round 3
Reviewer 1 Report
Comments and Suggestions for Authors
Thank you for taking my critique seriously. I believe the manuscript is much improved by the additional analysis and discussion of previous similar studies.